# Biomagnetism: The First Sixty Years

**DOI:** 10.3390/s23094218

**Published:** 2023-04-23

**Authors:** Bradley J. Roth

**Affiliations:** Department of Physics, Oakland University, Rochester, MI 48309, USA; roth@oakland.edu; Tel.: +1-248-375-2703

**Keywords:** biomagnetism, inverse problem, magnetocardiogram, magnetoencephalogram, optically pumped magnetometer, SQUID magnetometer

## Abstract

Biomagnetism is the measurement of the weak magnetic fields produced by nerves and muscle. The magnetic field of the heart—the magnetocardiogram (MCG)—is the largest biomagnetic signal generated by the body and was the first measured. Magnetic fields have been detected from isolated tissue, such as a peripheral nerve or cardiac muscle, and these studies have provided insights into the fundamental properties of biomagnetism. The magnetic field of the brain—the magnetoencephalogram (MEG)—has generated much interest and has potential clinical applications to epilepsy, migraine, and psychiatric disorders. The biomagnetic inverse problem, calculating the electrical sources inside the brain from magnetic field recordings made outside the head, is difficult, but several techniques have been introduced to solve it. Traditionally, biomagnetic fields are recorded using superconducting quantum interference device (SQUID) magnetometers, but recently, new sensors have been developed that allow magnetic measurements without the cryogenic technology required for SQUIDs.

## 1. Introduction

This year marks the 60th anniversary of the birth of biomagnetism: the study of magnetic fields produced by the body. Ultrasensitive sensors let researchers record the magnetic field of the heart and brain. Studies of isolated nerves and muscle have answered fundamental questions about how biomagnetic fields relate to their current sources. Innovative technologies may encourage new medical applications.

## 2. The Magnetocardiogram

### 2.1. The First Measurement of the Magnetocardiogram

In 1963, Gerhard Baule and Richard McFee first measured the magnetic field generated by the human body [1]. Working in a field in Syracuse, New York, they recorded the magnetic field of the heart: the magnetocardiogram (MCG). To sense the signal, they wound two million turns of wire around a dumbbell-shaped ferrite core that responded to the changing magnetic field by electromagnetic induction. The induced voltage in the pickup coil was detected with a low-noise amplifier.

The ferrite core was about one-third of a meter long, so the magnetic field was not measured at a single point above the chest, but instead was averaged over the entire coil. One question repeatedly examined in this review is spatial resolution. Small detectors are often noisy and large detectors integrate over the area, creating a trade-off between spatial resolution and the signal-to-noise ratio.

The heart’s magnetic field is tiny, on the order of 50–100 pT (Figure 1). A picotesla (pT) is less than a millionth of a millionth as strong as the magnetic field in a magnetic resonance imaging machine. The magnetic field of the earth is about 30,000,000 pT (Figure 1), and the only reason it does not obscure the heart’s field is that the earth’s field is static. That is not strictly true. The earth’s field varies slightly over time, which causes geomagnetic noise that tends to mask the magnetocardiogram (Figure 1). Moreover, even a perfectly static geomagnetic field would influence the MCG if the pickup coil slightly vibrated. A key challenge in biomagnetic recordings, and a major theme in this review, is the battle to lower the noise enough so the signal is detectable.

Baule and McFee suppressed background noise by subtracting the output of two pickup coils. A distant source of noise gave the same signal in both coils and did not contribute to their difference. One coil was placed over the heart, and the magnetocardiogram was larger there and did not cancel out. The two coils formed a rudimentary type of gradiometer (Figure 2).

Most laboratories contain stray magnetic fields from sources such as electronic equipment, elevators, or passing cars (Figure 1). Baule and McFee avoided much of this noise by performing their experiments at a remote location. Even so, they had to filter out the ubiquitous 60 Hz magnetic field arising from electrical power distribution. A magnetic field changing at 60 Hz is a particular nuisance for biomagnetism because the magnetic field typically exists in a frequency band extending from 1 Hz (1 s between heartbeats) to 1000 Hz (1 ms rise time of a nerve or muscle action potential). 

One limitation of a metal pickup coil is the thermal currents in the winding due to the random motion of electrons, creating extraneous magnetic fields caused by the measuring device itself. The ultimate source of noise is thermal currents in the body, but fortunately, their magnetic field is miniscule (Figure 1).

The magnetocardiogram resembled the electrocardiogram (ECG) sensed by electrodes attached to the skin. Baule and McFee speculated that the MCG might contain different information than the ECG, another idea that reappears throughout this review. In a follow-up article, they theoretically calculated the magnetic field produced by the heart [3]. The interplay between theory and experiments is yet one more subject that frequently arises in this article.

### 2.2. The Shielded Room and the SQUID

After Baule and McFee’s groundbreaking study, research into magnetocardiology was taken up by David Cohen, one of the founders of biomagnetism. Cohen performed his experiments inside a magnetically shielded room [4]. The walls of the enclosure were made out of two layers of 1.5 mm-thick molypermalloy, a metal having high magnetic permeability. It reduced the magnetic noise by a factor of a thousand, so he could observe the magnetocardiogram with a smaller pickup coil than Baule and McFee used (but still wound with hundreds of thousands of turns), with no need for a second coil to form a gradiometer. Shielded rooms are now common in biomagnetism laboratories, but they are expensive. Some are built from magnetic materials such as molypermalloy or mu-metal—an alloy containing nickel, iron, and molybdenum—to shield from all magnetic fields, including static ones. Others are constructed from aluminum, so the induced eddy currents in the metal walls block time-varying magnetic fields from penetrating into the room [5,6,7].

The voltage induced in Cohen’s pickup coil had a time course that was the derivative of the magnetic field, but he electronically integrated his signal before comparing the MCG to the ECG. In some cases, he averaged over 150 heartbeats to further lessen the noise.

In the 1960s, an instrument was invented to detect extraordinarily weak magnetic fields: the superconducting quantum interference device (SQUID) magnetometer (Figure 1). In 1970, Cohen—working in the Francis Bitter Magnet Laboratory at the Massachusetts Institute of Technology—teamed up with one of the inventors of the SQUID, James Zimmerman, to conduct the first biomagnetic measurements with this ingenious technology [8]. A SQUID must be kept at ultracold temperatures in order to remain superconducting (a few degrees above absolute zero), so it was placed within a thermally insulated dewar filled with liquid helium. SQUIDs provided smaller detectors with increased sensitivity compared to wire-wound pickup coils, at the cost of requiring cryogenic technology. By recording with a SQUID in a shielded room, Cohen and Zimmerman could detect the MCG with no averaging. Following Cohen’s lead, in subsequent decades, SQUID magnetometers became the state-of-the-art instrument for performing biomagnetic measurements [9,10].

### 2.3. Growth of Magnetocardiology in the 1970s

After their initial study, Cohen and Zimmerman continued to investigate the magnetocardiogram. Zimmerman and Nolan Frederick showed that by utilizing a first-order gradiometer design for the superconducting pickup coil (Figure 2), SQUID magnetometers could record the MCG without the need for a shielded room [11]. Cohen observed magnetocardiograms from normal patients [12] and patients with heart disorders [13], with the goal of eventually using the MCG as a clinical tool.

At the same time, other research teams became interested in the growing field of biomagnetism. One of the earliest and most successful groups was led by Toivo Katila at the Helsinki University of Technology (now part of Aalto University) in Finland. In 1974, they reported the first observation of a fetal magnetocardiogram, which allows a doctor to monitor the heart rate of a fetus [14]. They also began recording MCGs from people with cardiac arrhythmias, such as atrial fibrillation, bundle branch block, and premature ventricular contractions [15]. Other laboratories began studying the MCG, too [16,17], including one at Stanford University that measured all three components of the magnetic field and used these data to determine the “magnetic heart vector” [18]. In August 1976, the field had advanced to the point that the first Biomagnetism Conference was held in Cambridge, Massachusetts. Ever since, BIOMAG conferences have occurred biennially all over the world. The 23rd conference will take place during August of 2024 in Sydney, Australia (see: www.biomag2024.org).

Similar to the electrocardiogram, the magnetocardiogram consists of three deflections: a P wave caused by activation of the atria, a larger QRS complex from activation of the ventricles, and a T wave corresponding to recovery of the ventricles (Figure 3). In 1975, Cohen and his collaborator Larry Kaufman recorded MCGs from dogs who had undergone a coronary artery occlusion a week earlier [19]. The magnetic signal from each of the damaged hearts was elevated between the end of its QRS complex and the start of its T wave (called the “ST segment”). They used a SQUID magnetometer that could sense a steady (“direct current”) magnetic field, a dcSQUID; before that time, most SQUIDs had been sensitive only to fields that vary with time (“radio frequency”), rfSQUIDs. By measuring the MCG as they moved the dog toward or away from the dcSQUID, they demonstrated that during the ST segment, the signal was zero (shown by the dashed line in Figure 3), but the magnetic field at other times, such as after the end of the T wave until the onset of the following P wave, was not zero, and was caused by injury currents. Ischemia (a lack of oxygen) had caused some of the cardiac muscle to be injured and to remain depolarized (loss of the resting potential that normally exists across the muscle cell membrane). Current flowed from the injured tissue into the surrounding healthy resting tissue, resulting in a nonzero field. During the plateau between the QRS complex and the T wave, however, both the healthy and injured tissue were depolarized, eliminating the injury current. Gerhard Stroink and his colleagues at Dalhousie University in Halifax, Canada, showed that the detection of an elevated ST segment in magnetocardiograms of patients indicated that they had suffered from myocardial ischemia brought about by coronary artery disease (they had a heart attack) and might be at risk for a ventricular arrythmia [20]. Researchers in Finland came to similar conclusions [21], as did others [22,23]. The MCG has become an alternative to the ECG for diagnosing coronary artery disease. It has the advantage of being contactless and sensitive, but the disadvantage is that the instrumentation is expensive and complicated.

### 2.4. Theoretical Calculations of the Magnetocardiogram

David Geselowitz and, later, Robert Plonsey analyzed the relationship between the biomagnetic field and the current sources that gave rise to it [24,25]. They divided the source into two parts: primary and secondary. The primary source is the current crossing the cell membrane, called the impressed current density. Since the current that leaves the intracellular space at one location must reenter it at another to form a closed loop, the primary source is usually represented as a current dipole. Secondary sources arise at the boundaries between surrounding tissues having unequal conductivities. The most obvious boundary is that between the torso and air, but a boundary between different tissues also produces a secondary source. Double layers of charge build up at these boundaries and the resulting electric field creates secondary currents that contribute to the magnetic field. Geselowitz and Plonsey developed the integral equations that govern how these primary and secondary sources generate the magnetic field.

Milan Horáček, of Dalhousie University, implemented a computational method to solve these integral equations [26]. He represented each boundary with triangles, so the integrals over the boundary surfaces became a sum over 1884 triangles: 1216 triangles for the torso, 326 for the lungs, and 342 for the inner surface of the heart. He derived a linear system of 1884 equations that governed the secondary sources and solved them on a computer. His model allowed him to simulate the magnetocardiogram [26,27]. Horáček’s calculations were a harbinger of the extensive utilization of computers in biomagnetic analysis.

While the forward problem of determining the magnetic field from the current is important, of even greater interest is the inverse problem of determining the current from the measured magnetic field. Indeed, how to solve the inverse problem is the paramount question in biomagnetism. The simplest way to approach it is to assume the primary source consists of a single dipole. The forward problem can be solved to calculate the magnetic field it produces [26]. This calculated field is then compared to the measured field, and the least-squares method determines the dipole position, orientation, and strength that minimize the difference between them. Since the magnetocardiogram is a function of time, the best fit can be computed at every moment. In general, the dipole will be localized at different locations at each time, giving rise to the “moving dipole method” [28], which has been used to identify the ventricular preexcitation site in patients with Wolff–Parkinson–White syndrome [29].

### 2.5. Information Content of the Magnetocardiogram

A vital question is: does the magnetocardiogram contain information not present in the electrocardiogram? Plonsey pointed out that the ECG is produced by the divergence of the impressed current density, while the MCG is produced by its curl [30]. Since these two sources could be unrelated, the MCG and ECG might contain independent information. Stanley Rush, on the other hand, argued that physical constraints require the divergence and curl to be related, so that the MCG and ECG are “fundamentally interdependent” [31]. John Wikswo and John Barach, at Vanderbilt University, countered Rush’s claim by noting that cardiac muscle is anisotropic, so the impressed current could have a component that is not perpendicular to the voltage gradient associated with the wave front [32]. As an example, consider an action potential propagating from the inner surface of the left ventricle of the heart to the outer surface. Figure 4 shows a cross-section of the ventricle, with the depolarized myocardium in dark pink and resting tissue in light pink. In the isotropic case (Figure 4, left), the impressed current, and consequently the dipole source (green arrows in Figure 4), point radially out. In anisotropic tissue, however, the impressed current could have a tangential component forming a current loop circling around the ventricle wall, an “electrically silent magnetic source” (Figure 4, right). Both cases in Figure 4 would give rise to the same ECG, but they would produce different MCGs. Wikswo and Barach concluded that “it is possible for the MCG to contain information not present in surface ECG measurements” [32].

### 2.6. Instrumentation and Clinical Applications

Many of the early recordings of the MCG were performed with a single SQUID sensor. To determine the spatial distribution of the magnetic field, an experiment had to be repeated many times, each time with the sensor in a new position. Beginning around 1990, multichannel systems became available that allowed scanning at numerous locations simultaneously. One of the first such devices had seven channels [33], but soon the number increased [34], until over a hundred channels were available [35]. These systems permitted the study of arrhythmias that were not periodic and dramatically shortened the time required to make a magnetic field map. Furthermore, once commercial SQUID systems could be purchased off-the-shelf rather than being designed and manufactured in-house, groups without cryogenic expertise were able to obtain clinical MCG data. 

Riccardo Fenici and his colleagues have reviewed the medical applications of magnetocardiography [36], including diagnosing myocardial ischemia [23], assessing the risk of an arrhythmia [37], imaging of arrhythmogenic mechanisms, measuring the fetal heartbeat [38], and possibly even detecting left ventricular overload and hypertrophy. While the data analysis for all these applications was originally performed using traditional methods, recent advances in artificial intelligence and machine learning have permitted a shift toward automatic MCG analysis to facilitate more rapid and accurate clinical diagnosis [39,40,41].

## 3. Isolated Tissue

### 3.1. Magnetic Field of a Nerve

While much of the early biomagnetic research centered on measuring the magnetocardiogram, Wikswo took a different track—he sought to measure the magnetic field produced by isolated tissue. In 1980, Wikswo, Barach, and John Freeman dissected a sciatic nerve out of a frog, threaded it through a wire-wound ferrite-core toroid, immersed the toroid in a saline bath, and then electrically stimulated the nerve [42]. An action potential propagated along the nerve, producing action currents (Figure 5). The intracellular current associated with the depolarization (upstroke) of the action potential points to the left in Figure 5, and the current associated with the repolarization (downstroke) points to the right. The current in the bath completes each loop. A magnetic field associated with these currents encircles the nerve. As the action potential propagates to the left, a magnetic field is created in the toroid, first in one direction and then in the other. This changing magnetic field induces a voltage in the winding.

The magnetic field sensed by the toroid (inner radius 0.6 mm, outer radius 1.3 mm, thickness 1.2 mm) was 70 pT, which is similar to the magnetic field of the magnetocardiogram. When the saline bath was removed, so merely a thin layer of fluid clung to the nerve’s outer surface, the magnetic field disappeared. If the action potential propagated in the opposite direction, the magnetic field changed polarity. In their original study, Wikswo et al. detected the current in the toroid coil with a SQUID magnetometer. Later, they found that the extreme sensitivity of a SQUID was not needed, and they developed a room-temperature, low-noise amplifier for subsequent experiments [43]. 

Toroids have been used in a variety of experiments to measure the magnetic field from isolated tissue. One of these experiments was to measure the field of a single nerve axon. Wikswo and his graduate student Brad Roth simultaneously recorded the magnetic field (using a toroid) and transmembrane potential (using a glass microelectrode) of a crayfish giant axon having a radius of 0.1 mm [44]. The rate of change of the induced magnetic signal sensed by a toroid was sent to an integrator to obtain the magnetic field itself [45]. The peak magnetic field was approximately 200 pT. Toroids have also been used to analyze compound action currents in a nerve [46,47,48] and a single skeletal muscle fiber [49]. They were even used to assess an axon damaged at its end [50]. A limitation of this technique is that the nerve needs to be cut and threaded through the toroid, but Mark Leifer and Wikswo developed an openable toroid that avoided any need for dissection [51]. Toroids have been used to evaluate the continuity of a nerve during surgery [52,53].

### 3.2. Ampere’s Law Versus the Law of Biot and Savart

The magnetic field of an axon can be calculated using either Ampere’s law or the law of Biot and Savart [54,55,56]. Ampere’s law states that the integral of the magnetic field along a closed loop is proportional to the net current threading the loop. If the geometry has sufficient symmetry, the line integral becomes simply the magnetic field multiplied by the loop’s circumference. The current threading the loop consists of two parts: the intracellular current and that fraction of the extracellular current that returns through the loop, called the “return current” [44]. If the toroid fits snug around the axon, most extracellular current circles back outside of it, so the return current is small. In that case, the current threading the loop is equal to only the current inside the axon. In a larger toroid, however, more extracellular current returns through it, partially cancelling the intracellular current.

The law of Biot and Savart, on the other hand, integrates the intracellular current along the length of the axon. The extracellular current contributes little or nothing because it is the sum of the current coming out of the axon at different points, and the magnetic field from the current radiating outward from a point is zero. The magnetic field of an axon is not the same as for a long, straight wire because the action potential has a finite length. The depolarization current points opposite to the repolarization current, so they partially cancel each other in the integral.

Wikswo and Roth, working with Jim Woosley, performed these integrals using Fourier transforms and modified Bessel functions [55]. The mathematical expression for the magnetic field they calculated from Ampere’s law did not look the same as that calculated from the law of Biot and Savart. However, by employing an esoteric relationship among Bessel functions, they were able to show that these expressions were indeed equivalent, so the magnetic field was identical whether derived from Ampere’s law or the law of Biot and Savart. A discrepancy arose only if they tried to assign different parts of the magnetic field to the intracellular or extracellular current. Using the law of Biot and Savart, only the intracellular current significantly contributed to the magnetic field and not the extracellular current. Using Ampere’s law, intracellular and extracellular currents were both significant, especially when the magnetic field was evaluated far from the axon. The total magnetic field—the field that you measure experimentally—was the same in the two cases, but if you designate one part of the field as arising from an intracellular source and another part from an extracellular source, you obtain contrasting physical pictures from the two laws.

For measurements of the nerve magnetic field with a toroid, the interpretation based on Ampere’s law is particularly useful. For instance, the reason the magnetic field was zero when Wikswo removed the saline bath was that the net current threading the toroid vanished: the intracellular current was canceled by an equal but opposite extracellular current passing through the thin layer of saline on the nerve surface [42]. Nevertheless, the interpretation based on the law of Biot and Savart is used most frequently in biomagnetism, such as when analyzing the magnetocardiogram. It easily lends itself to a multipole expansion of the current source [57,58]. For example, the electrical activity of the heart is represented as a current dipole, and an action potential propagating down a nerve produces two equal but opposite dipoles, resulting in quadrupole behavior.

### 3.3. Magnetic Field Produced by Cardiac Muscle

Wikswo’s laboratory studied other tissue besides nerves. Roth and Marcella Woods examined an action potential propagating through a two-dimensional sheet of cardiac muscle [59]. In Figure 6, a wave front is propagating to the right, so the myocardium on the left is fully depolarized and on the right is at rest. Cardiac muscle is anisotropic, meaning it has a different electrical conductivity parallel to the myocardial fibers than perpendicular to them. In Figure 6, the fibers are oriented at an angle to the direction of propagation. The intracellular voltage gradient is in the propagation direction (horizontal in Figure 6), but the anisotropy rotates the intracellular current toward the fiber axis. The same thing happens to the extracellular current, except that in cardiac muscle the intracellular conductivity is more anisotropic than the extracellular conductivity, so the extracellular current is not rotated as far. Continuity requires that the components of the intracellular and extracellular current densities in the propagation direction are equal and opposite. Their sum, therefore, points perpendicular to the direction of propagation, creating a magnetic field that comes out of the plane of the tissue on the left and into the plane on the right (Figure 6) [59,60,61]. This perspective of the current and magnetic field in cardiac muscle is unlike that ordinarily adopted when analyzing the magnetocardiogram, where the impressed current is typically taken in the same direction as propagation. Nonetheless, experiments by Jenny Holzer in Wikswo’s lab confirmed the behavior shown in Figure 6 [62].

An illuminating case study is propagation originating at the apex of the heart [63]. Suppose part of the heart wall at the apex is dissected out, forming a slab, as shown in Figure 7 (top panel). This slab is super-fused above and below by saline. Myocardial fibers have a distinctive geometry near the apex: they spiral outward. Assume an electrical stimulus is applied at the apex, triggering a wave front that propagates radially outward. Two mechanisms are responsible for the magnetic field [64]. First, a situation analogous to that for a nerve develops, where the intracellular current points outward in the direction of propagation and the extracellular current returns in the saline bath (Figure 7, left panel; compare to Figure 5). The resulting magnetic field circulates in planes above and below the slab. This is the traditional view of how a cardiac magnetic field is produced. Second, intracellular and extracellular anisotropy generates a loop of current rotating around the apex (Figure 7, right panel; compare to Figure 6). The resulting magnetic field resembles that formed by a coil of wire and has a component perpendicular to the slab. Calculations imply that both mechanisms operate simultaneously, and the total magnetic field has a spiraling pattern, as shown in the bottom panel of Figure 7. As with the example in Figure 6, this theoretical prediction has been verified in Wikswo’s laboratory by Krista McBride [65]. It supports the hypothesis presented in Figure 4 that the magnetocardiogram contains information not existing in the electrocardiogram.

### 3.4. MicroSQUID

The experiments confirming the predictions in Figure 6 and Figure 7 required the development of a “SQUID microscope,” named microSQUID [66]. SQUID magnetometers sense the magnetic field and must be kept cold in liquid helium. In contrast to traditional devices, in microSQUID the superconducting pickup coils were maintained in the vacuum space between the room-temperature sample and the ultra-low-temperature liquid helium. This design allowed the coils to be placed a mere 0.1 mm above the sample, providing exceptional spatial resolution. This technology was the basis for several studies of cardiac muscle [62,65,67,68].

In order to interpret microSQUID data, Roth and Wikswo, working with Nestor Sepulveda, derived a general mathematical technique to determine the two-dimensional current distribution from the measurement of its magnetic field [69]. The process, which makes use of Fourier transforms, provides a unique solution to the inverse problem. In general, the biomagnetic inverse problem has no unique solution, but if the current is restricted to a two-dimensional plane, it can be completely solved. The Fourier method provides insight into why the inverse problem is so difficult [69,70]. The lower spatial frequencies are nicely recovered in the image of the current. Higher spatial frequencies in the current distribution, however, insignificantly contribute to the measured magnetic field, which makes the inversion process exquisitely sensitive to any high spatial frequency noise. Low-pass spatial filtering allowed the reconstruction of a reasonable, if smeared out, image of the current. 

Investigations of isolated nerves and muscle provide insight into many basic concepts underlying biomagnetism. However, nowadays, the field is driven by clinical applications, especially for the study of the brain.

## 4. The Magnetoencephalogram

### 4.1. First Measurements of the Magnetoencephalogram

In 1968, biomagnetism pioneer David Cohen performed the first measurement of the magnetic field of the brain: the magnetoencephalogram (MEG). He detected the brain’s largest signal: the alpha rhythm [71]. This nearly sinusoidal oscillation at a frequency of about 10 Hz is turned on or off by closing or opening your eyes. Since this experiment was performed before SQUID magnetometers were introduced into biomagnetism, Cohen wound a million-turn pickup coil to record the MEG. He averaged 2500 times by triggering off the electroencephalogram (EEG). Subsequent studies used a SQUID magnetometer to sense the approximately 1 pT magnetic field (Figure 1) [72].

The MEG is primarily produced by currents in the dendrites of neurons in the brain [73,74]. Magnetic fields evoked in these neurons by a stimulus are about ten times weaker than those associated with the alpha rhythm and are a thousand times weaker than the magnetocardiogram (Figure 1). The visual-evoked magnetic field was detected by Sam Williamson and his colleagues in 1975 [75]. By using a second-order gradiometer (Figure 2), they were able to record a 0.1 pT signal in their laboratory in downtown Manhattan, with no shielded room. Similar results were obtained by Cohen’s group [76] and others [77]. Soon, evoked magnetic fields were observed during electrical stimulation of the finger [78], following voluntary finger flexion [79,80], and in response to a sound [81,82,83,84,85,86]. MEG recordings were sensitive enough to demonstrate how distinct regions in the brain responded to different sound frequencies: tonotopic organization [87]. 

### 4.2. Magnetic Field of a Dipole in a Spherical Conductor

The initial calculations of the magnetic field underlying the magnetoencephalogram were performed assuming that the head is a sphere [88,89,90]. An influential article by Jukka Sarvas, from the biomagnetism group in Finland, summarized three essential facts about the magnetic field of a dipole in a spherical conductor [91], as follows:A radial dipole produces no magnetic field (Figure 8). This result is best proven using Ampere’s law: the magnetic field integrated along a closed loop is proportional to the net current threading the loop. The symmetry is sufficient that the integral over the path (dashed circle in Figure 8) equals the path length times the magnetic field. The current produced by a dipole, including the return current, must be contained within the sphere because the region outside is not conducting. Hence, the net current threading the loop (dipole plus return current) is zero, so the magnetic field of a radial dipole vanishes.The radial component of the magnetic field is the same as for a dipole in a homogenous, unbounded conductor. In other words, the conductor–insulator interface at the surface of the sphere does not affect the radial component of the magnetic field. Furthermore, this result holds for a heterogeneous sphere, as long as the conductivity varies only with depth. For these reasons, investigators prefer measuring the radial component of the magnetic field when studying the MEG.The tangential component of the magnetic field is affected by the conductor–insulator boundary, but it does not depend on a conductivity that radially varies.

These results have a significance for the magnetoencephalogram. You cannot record a magnetic field from radial dipoles, such as those that may exist in cortical folds or gyri [92]—they are magnetically silent. The electroencephalogram does have a signal from a radial dipole, so an advantage of the EEG over MEG is that the EEG detects radial sources. The three-sphere model—consisting of concentric spheres representing the brain, skull, and scalp—is often used to model the head. The magnetoencephalogram is independent of a conductivity that radially varies, so it does not depend on the low conductivity of the skull. The electroencephalogram is dramatically influenced by the skull conductivity, so an advantage of the MEG over the EEG is that the MEG is independent of the skull, scalp, and brain conductivities. Numerical calculations [93,94,95] and phantom studies [96,97,98,99,100] show that these conclusions that were rigorously proven for a sphere are good, but not perfect, approximations for a realistically shaped head.

### 4.3. Relative Merits of the Magnetoencephalogram and the Electroencephalogram

The last section analyzed some of the relative merits of the magnetoencephalogram and the electroencephalogram. Are there enough advantages of the MEG over the EEG to justify the cost and complexity of cryogenics? The magnetic field pattern is rotated by 90° relative to the electrical voltage distribution, so the MEG should provide better localization perpendicular to a dipole, while the EEG should better localize parallel to it. In addition, the MEG might reveal tangential dipoles that are obscured in the EEG by radial ones [101]. 

In 1990, Cohen created a stir by claiming that the MEG did not localize a source better than EEG did [102]. He and his coworkers implanted dipoles at known locations in the brains of patients undergoing seizure monitoring. When they solved the inverse problem to determine the location of the dipole from either electric or magnetic data, they found each had a localization error of about 10 mm. Critics such as Sam Williamson [103] and a team led by Riitta Hari [104] claimed that Cohen’s study was limited by obsolete equipment and poor procedures. Wikswo, Williamson, and Alan Gevins weighed the pros and cons of the two techniques [105]. Measuring the EEG is less expensive than the MEG and does not require as extensive shielding. The patient is able to move while recording the EEG because the electrodes are attached to the scalp, but the head has to be motionless when measuring the MEG. The MEG is less affected than the EEG if you pick a wrong value of the skull conductivity when solving the inverse problem.

By the mid-1990s, the MEG was becoming an established technique for investigating brain activity. In 1993, Matti Hämäläinen and his collaborators in Finland surveyed magnetoencephalography in the *Reviews of Modern Physics* [2]. It is the most highly cited publication in the field of biomagnetism and continues to provide one of the best summaries of MEG theory, instrumentation, and applications.

### 4.4. Solving the Inverse Problem

How to solve the inverse problem—determining the sources of brain activity from magnetic measurements—is the central question in magnetoencephalography. In general, the solution is not unique. For example, if you assume a spherical head, then you may always add a radial dipole to any solution [91]. In addition, the inverse solution is often not stable. Slight measurement errors propagate through an algorithm to cause major errors in the calculated sources. In other words, the problem is ill-posed, which means that “trusting the data too much will unquestionably lead to nonsense solutions” [106].

The inverse problem was initially solved for a single dipole. Its location, orientation, and strength were determined by a least-squares fit to the experimental data. The technique can be generalized to multiple dipoles [107,108], but with too many the algorithm becomes underdetermined: the number of source parameters exceeds the number of data points. In this case, numerous solutions might provide equally good fits and a method is needed to select the optimal one. A common method is to select the minimum-norm solution: the solution having the least magnitude squared when summed over all dipoles. The least-squares minimum-norm solution is found using singular-value decomposition [109,110].

Often, least-squares minimum-norm methods are applied at one instant. If solved at different times, the resulting dipoles could move or rotate. An alternative approach is to assume a few fixed dipoles that each produce their own individual (asynchronous) time series. The fit is performed over their postulated locations and time (spatial-temporal analysis). John Mosher and Richard Leahy have applied this multiple-signal classification (MUSIC) algorithm, and a recursively applied and projected MUSIC algorithm (RAP-MUSIC), to MEG data [111,112,113].

The MUSIC algorithm has been generalized by employing spatial filtering (beamformers), where the filter emphasizes regions of interest within the brain while attenuating other locations [114,115]. Beamformers are best-suited for studying processes that are coherent in time but are spatially segregated. An advantage of this algorithm is that you do not have to make any assumption about the number of sources. 

Still another approach to solving the MEG inverse problem is to use known anatomical and physiological constraints. The least-squares minimum-norm fit is performed subject to these constraints by adopting a Bayesian formulation to produce a “maximum a posteriori” estimate of brain activity [106]. One implementation of this technique is known as low-resolution electromagnetic tomography (LORETA) [106,116,117]. Such procedures require that the MEG data be integrated with anatomical information from other methods, such as MRI. 

If you have no anatomical information, or you do not wish to form any hypotheses about the source, you can go to the other extreme and construct a solution that directly relates the magnetic field to the current, bypassing any restrictive assumptions or least-squares fit. An example was described in Section 3.4, where Fourier analysis determined the current in a two-dimensional sheet from the magnetic field observed in a plane above the sheet [69]. Similarly, Hämäläinen and Risto Ilmoniemi related the current in the brain to the magnetic field measured outside a spherical head, reducing the inverse problem to a linear estimation [118]. To keep their problem well-posed, they had to apply a minimum-norm constraint and a regularization procedure. This technique predicts current distributions that explain the MEG data with hardly any a priori assumptions [119,120]. The solutions tend to be diffuse, and the procedure often poorly localizes restricted sources. 

A variation of linear estimation for the inverse problem is the focal underdetermined system solution (FOCUSS) [121]. The algorithm starts with a minimum-norm solution and then recursively strengthens the sources in some regions and suppresses them in others, until they exist in only a small number of locations. The weightings that determine which regions are strengthened or suppressed are found from previous iterations, so the algorithm tends to focus on a handful of sources. This algorithm is particularly useful when there is reason to suspect that the MEG is produced by a few, well-localized sites. Norman Tepley and his team have developed a multi-resolution version of FOCUSS that uses wavelets to make the algorithm less susceptible to noise [122,123].

This review cannot examine all the various algorithms for solving the inverse problem, there are just too many. For example, whole families of methods are based on coherence [124] and connectivity [125,126]. One of the most promising ways to contribute to MEG research is to further develop novel solutions to the inverse problem. It remains an important barrier limiting the use of MEG in medicine. 

### 4.5. Clinical Applications

Magnetoencephalography is able to diagnose and help treat many illnesses. Daniel Barth and his coworkers at UCLA used MEG to localize interictal spikes: transient, brief discharges, often observed in patients with epilepsy [127]. Such spikes occur in the same tissue as seizures originate from, so localization of the source of interictal spikes assists surgeons who perform brain surgery to treat epilepsy [128]. A clinical study of over 1000 participants provided some of the most impressive evidence yet for the utility of MEG in epilepsy diagnosis. It indicated that MEG “provides non-redundant information, which significantly contributes to patient selection, focus localization and ultimately long-term seizure freedom after epilepsy surgery” [129]. 

In addition to epilepsy, magnetoencephalography has other medical applications [130]. It has proven useful when diagnosing patients with impaired visual word processing due to dyslexia [131]. The data suggest that dyslexics have difficulty with written words because the parts of the left temporal lobe responsible for auditory language are impaired. Patients with Parkinson’s disease suffer from tremor, and MEG data indicate that involuntary activation of what is normally voluntary motor activity may trigger the tremor [132]. Direct current MEG demonstrates that depression-like activity spreads during migraine aura, which may cause the occipital cortex to become hyperexcitable [133]. Recordings of spontaneous MEG activity suggest that depression, tinnitus, neurogenic pain, and Parkinson’s disease may all be caused by a dysrhythmia in the interaction of the brain’s thalamus and cortex [134]. Even deep brain structures such as the hippocampus and amygdala can be monitored with MEG [135]. Sylvain Baillet of McGill University has reviewed the applications of MEG, and he expects it “to play an increasing and pivotal role in the elucidation of these grand mechanistic principles of cognitive, systems and clinical neuroscience” [136].

One way to encourage medical doctors to adopt MEG techniques is to make them easier and cheaper. SQUIDs constructed using high-temperature superconductors may allow the coolant to be liquid nitrogen instead of the more expensive and difficult to handle liquid helium [137,138]. Enormous shielded rooms sometimes might be replaced by active shielding, where the current passed through external coils cancels out extraneous magnetic field noise [9]. Software such as “brainstorm” [139] and “MNE” [140] is facilitating the use of MEG by clinicians. Artificial intelligence and deep learning are increasingly assisting doctors in interpreting the MEG [141].

Both the MEG and EEG must compete with other methods for imaging the brain, such as functional magnetic resonance imaging (fMRI) and positron emission tomography (PET). MEG and EEG provide better temporal resolution than fMRI and PET, but fMRI and PET provide better spatial resolution and do not require solving an ill-posed inverse problem. Although MEG is more expensive than EEG, it is not as expensive as fMRI or PET. Traditional MEG, similar to fMRI and PET, is completely non-contact (EEG requires attaching electrodes to the head, which may be difficult in some cases such as when there is damage to the scalp). MEG and EEG do not require injecting any radioactive isotopes into the patient, as PET does. fMRI measures blood flow in the brain and PET measures metabolic activity, whereas MEG and EEG directly record brain electrical activity. Ultimately, combinations of these techniques may be used together to learn more about how the brain works.

## 5. New Technologies

### 5.1. Optically Pumped Alkali Vapor Magnetometers

One of the most remarkable developments in biomagnetism over the last couple of decades is the introduction of new ways to detect the magnetic field, such as optically pumped magnetometers [142,143,144,145]. These instruments use a vapor of an alkali metal such as rubidium that is excited, or pumped, by circularly polarized light, and becomes magnetized. The magnetization precesses at its Larmor frequency, which means the vapor is affected by the local magnetic field. The magnetization is then probed by a second pulse of light. Optically pumped magnetometers operate at room temperature, so they do not require liquid helium like SQUIDs do. The original applications of these novel devices to MCG and MEG used a big, bulky volume of vapor, but the size of the detectors has been reduced to about 10 mm [146]. The vapor cell needs to be heated, but they can be thermally insulated from the scalp. Arrays of these detectors are commercially available and have been incorporated into a wearable cap [147,148]. 

Optically pumped magnetometers have advantages and disadvantages compared to traditional devices for measuring biomagnetic fields [149]. The magnetic field sensitivity of an optically pumped magnetometer is comparable to that of a SQUID. Since no dewar is needed to keep the detectors cold, an optically pumped magnetometer can be placed closer to the head than a SQUID can, almost on the scalp where the magnetic field is stronger, and the recording has more spatial resolution (especially for shallow sources) [150]. Unlike superconducting pickup coils, the optically pumped magnetometers are not easily configured as gradiometers, so they are susceptible to interference from external magnetic fields. As a result, most systems require magnetic shielding, comparable to that often needed by SQUIDs. Arrays of SQUID detectors are enclosed in a dewar so they cannot be adjusted to the contours of the patient’s head: they are one-size-fits-all systems that sometimes do not fit everyone, particularly children whose heads are little. The optically pumped magnetometers are attached to the scalp using a 3D-printed cast that conforms to the size of the head [147]. SQUID systems do not move as the head moves, so the patient should be still during the measurement. Optically pumped magnetometer arrays move with the head so they can be used to study people who are in motion. Any movement in an external magnetic field, however, would interfere with the recording, making magnetic shielding all the more crucial, and thus limiting the motion to within the shield. A wearable array of optically pumped magnetometers has been used to detect brain activity in children. The “on-scalp magnetoencephalography (MEG) based on optically pumped magnetometers helped detect interictal epileptiform discharges in school-aged children with epilepsy with a higher amplitude, higher signal-to-noise ratio, and similar localization value compared with conventional cryogenic MEG” [151]. 

### 5.2. Other Magnetometer Technologies

Another type of magnetic field detector contains nitrogen-vacancy defects in a diamond, which creates a spin system sensitive to a magnetic field that can be optically detected with a photodiode [152,153]. These detectors can be made smaller than those using alkali metal vapors but are less sensitive to the magnetic field. The detectors work best when you place them close enough to an isolated nerve or muscle, so the magnetic field is hundreds of picotesla in strength. Although the technology may improve in the future, these detectors do not appear to have sufficient sensitivity to compete with SQUIDs for measurement of the MEG outside the head.

Another type of magnetic field probe is based on giant magnetoresistance [154,155]. The measurement is simple, and no light is needed to probe the magnetic field. The devices are low-cost, flexible, and can be miniaturized, but are limited to detecting fields on the order of 1000 pT. They work well on the tip of a probe that is inserted into the brain of an animal, where the biomagnetic fields are large [156,157]. They have been used to record evoked MEG signals on the scalp, but extensive averaging was required [158].

Recently, magnetic sensors constructed from layered ferrite/piezoelectric composites have been proposed for measuring biomagnetic fields [159]. High-tech tools such as these may trigger more clinical applications of the MCG and MEG.

## 6. Conclusions

Biomagnetism remains a growing and developing field of study. New instruments may mean that biomagnetic measurements no longer require the complicated cryogenic technology associated with SQUID magnetometers. If these sensors can be arranged as gradiometers, or if active shielding can cancel out magnetic noise, then the need for expensive shielded rooms might be eliminated. In that case, the cost and complexity of devices to measure the MEG should become competitive with EEG machines, encouraging further medical applications. The next sixty years of biomagnetism might well be more momentous than the first sixty.

## Figures and Tables

**Figure 1 sensors-23-04218-f001:**
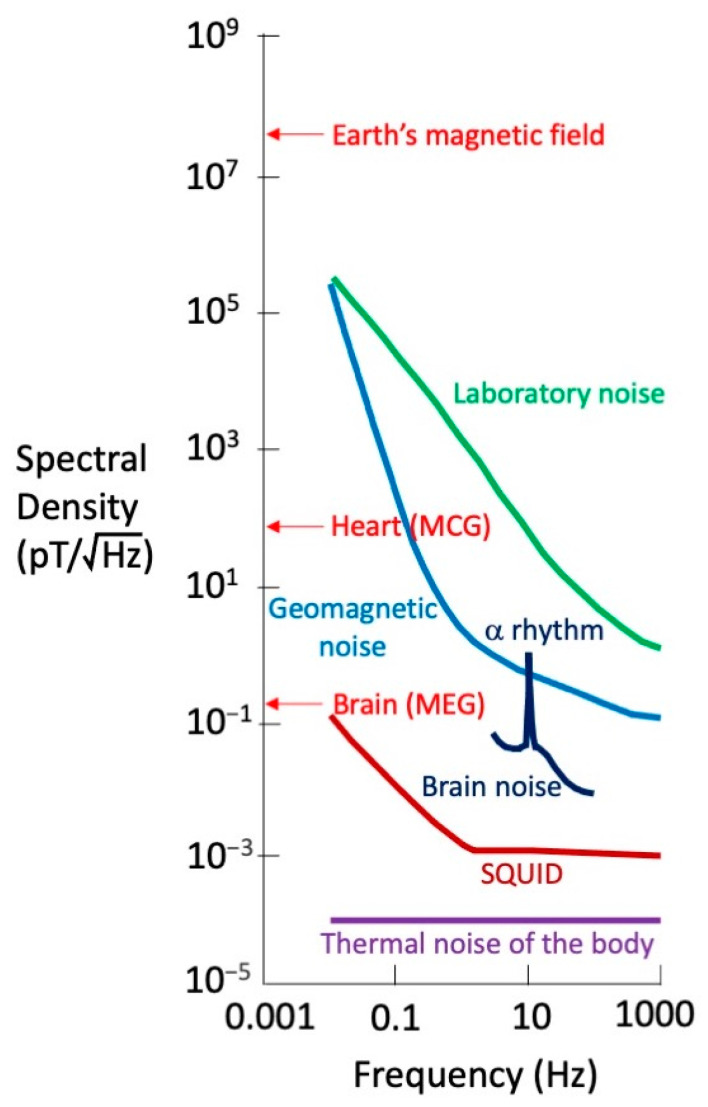
Noise sources in biomagnetism. The magnetic fields of the earth, heart, and brain are in pT, while the noise and SQUID sensitivity depend on frequency and are expressed in pT per root hertz. Reprinted with permission from [2]. Copyright 1993 by the American Physical Society.

**Figure 2 sensors-23-04218-f002:**
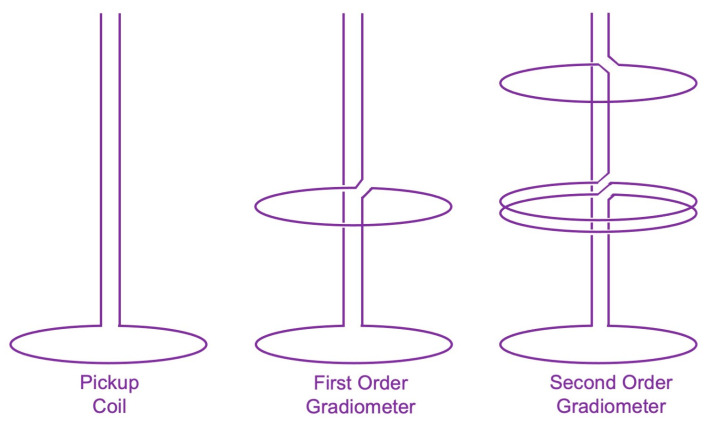
Types of gradiometers.

**Figure 3 sensors-23-04218-f003:**
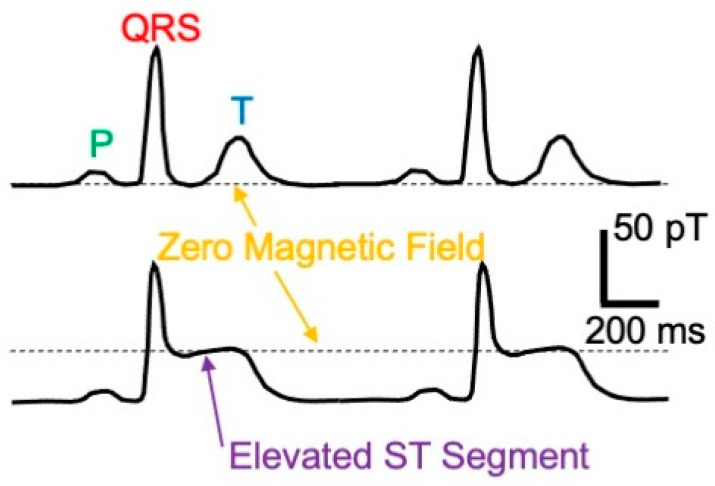
Top: Normal MCG. Bottom: MCG with an elevated ST segment.

**Figure 4 sensors-23-04218-f004:**
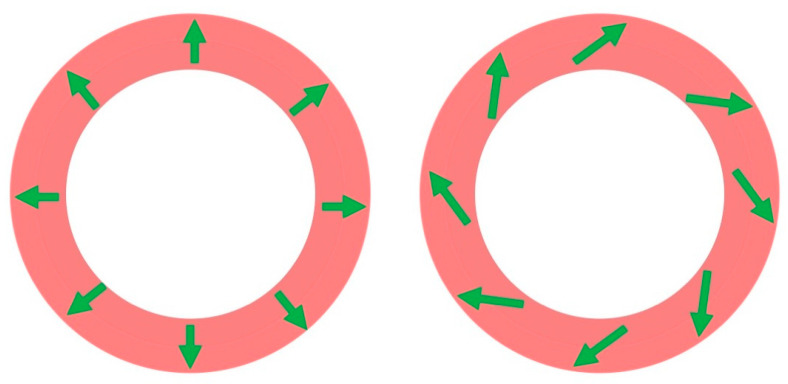
A cross-sectional view of the left ventricle. (**Left**): An isotropic tissue. (**Right**): Anisotropic tissue that produces a current loop around the ventricle wall.

**Figure 5 sensors-23-04218-f005:**
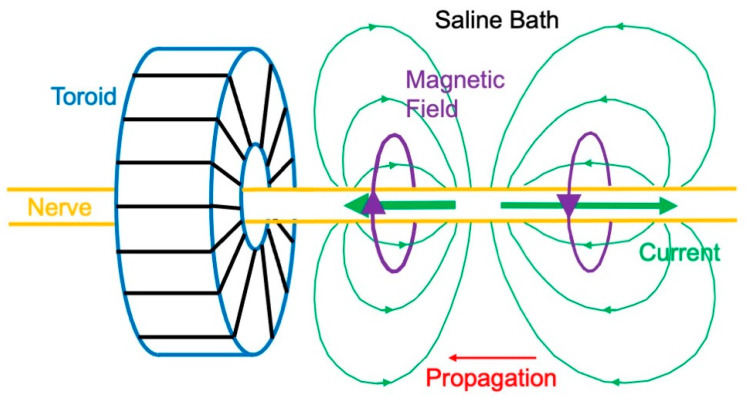
The magnetic field of a frog sciatic nerve, as measured using a toroid.

**Figure 6 sensors-23-04218-f006:**
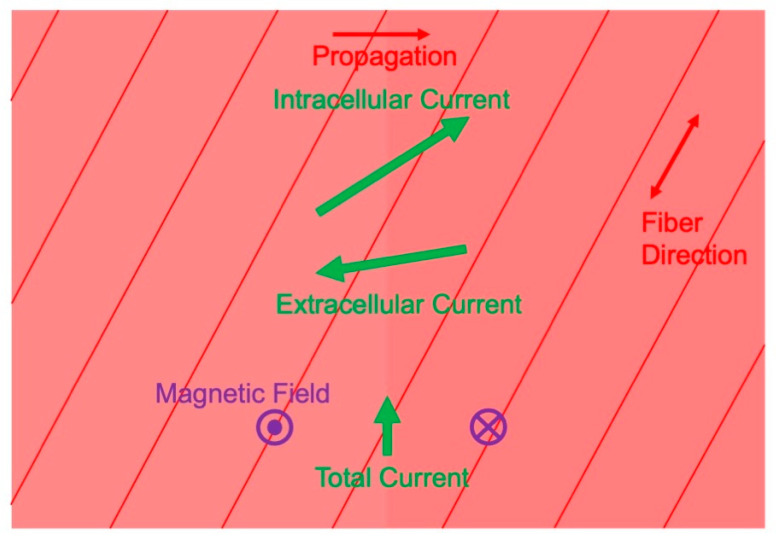
The current and magnetic field produced by a planar wave front propagating in a two-dimensional sheet of cardiac muscle. The muscle is anisotropic, with a higher conductivity along the myocardial fibers. Reproduced with permission from [59], copyright 1999 by the IEEE.

**Figure 7 sensors-23-04218-f007:**
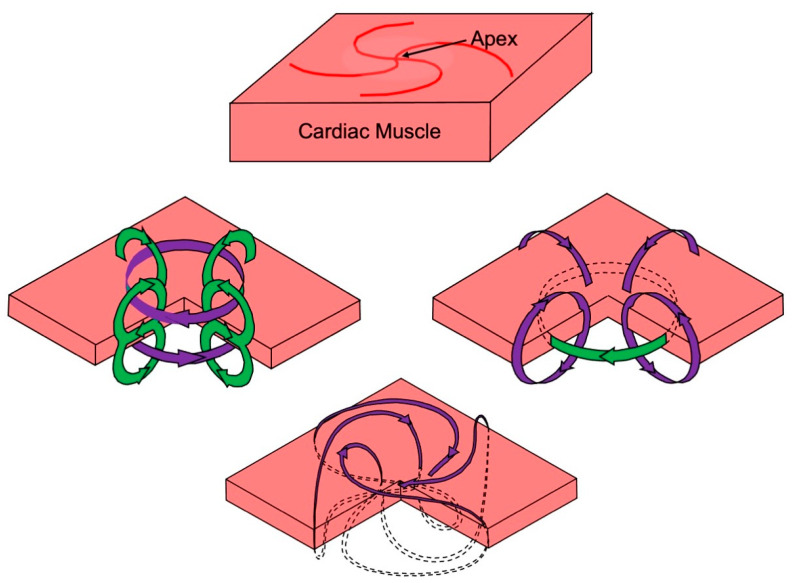
The magnetic field at the apex of the heart. Current is green and the magnetic field is purple. Reproduced with permission from [63], copyright 1988 by Elsevier.

**Figure 8 sensors-23-04218-f008:**
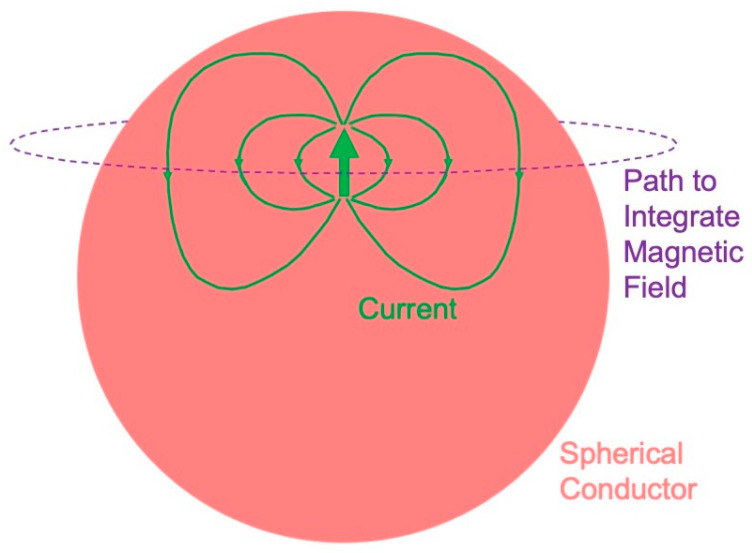
The magnetic field of a radial dipole is zero outside a spherical conductor.

## Data Availability

Not applicable.

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
