# Peer review of "Biomagnetism: The First Sixty Years"

_sensors, 2023, doi:10.3390/s23094218_

Round 1

Reviewer 1 Report

In this article, the author summarizes the first 60 years of the new scientific discipline of biomagnetism. The author chronologically and comprehensibly summarizes, with the help of citations, the measurement methods for obtaining the magnetocardiogram and magnetoencephalogram, as well as how the inversion methods were proposed. The article mentions clinical applications as well as new and promising magnetic sensors that can be used for measurements in the field of biomagnetism in the future. In my opinion, the article is well written and needs no further editing and I suggest accepting it as it is.

Author Response

Reviewer 1:

Thank you for your comments. I’m glad you liked the review.

Reviewer 2 Report

The author presents a summary of biomagnetism over the past 60 years. The summary is comprehensive and goes over the history, technical development, and their applications. 

Paragraph structure and sentence structure should be improved to facilitate flow and comprehension. 

In many instances, the paragraph starts talking about one topic and towards the end switches to another subject. The transitions could be made smoother so as to facilitate reading and avoid making the reader have to go back to previous paragraphs to fill in the gaps.    For example: section 2.6 P1 (lines 215-223): talks about the various setups for MCG recordings. Last sentence then states that once systems became commercially available, "groups without cryogenic expertise were able to obtain clinical MCG data". A naive reader would ask: why is that? is this because such systems did not use cryogenic technology? or is it because those systems did not require the same level of maintenance - and hence knowledge- from users? 
  Similarly, section 2.6 P2 (lines 224-229): talks about applications MCG but in the last sentence states that recently AI and ML have been used. This implies that the previously described analysis was manual in nature. The reader is also left wondering why this statement was made: why do we want to know that AI is now playing a role in data analysis? I'd recommend adding a sentence stating something like: "While the data analyses for all of these medical applications was originally done manually, recent advances in AI and ML have permitted a shift towards automatic MCG analysis to facilitate more rapid and accurate clinical diagnosis."    Section 3.1 P3 (line 256-267): starts by talking about the purpose of experiments in measuring the magnetic field. But then switches the subjects to Toroids and their applications. The paragraph needs to be reorganized and a thesis sentence needs to be added. It could be rewritten as follows, for example: "Toroids have been used in a variety of experiments to measure the magnetic field from tissue." example 1, example 2, example 3.    Section 3.2 last paragraph (line 302-311): talks about interpretation based on ampere's law. Then the last sentence talks about the electrical activity of the heart as a current dipole. A more obvious transition needs to be added so that the reader is not left wondering why we're talking about the heart or current dipoles. Transition phrases could help facilitate this "For example, in the case of...".    Several other paragraphs could benefit from having a thesis sentence at the beginning and more obvious transitions between sentences to facilitate reading and comprehension. 

Only minor editing is required (e.g unnecessary use of “the”). 

Author Response

Reviewer 2:

I thank the reviewers for their comments, which have improved the manuscript. All changes in the manuscript are marked in red.

“The author presents a summary of biomagnetism over the past 60 years. The summary is comprehensive and goes over the history, technical development, and their applications. 

Paragraph structure and sentence structure should be improved to facilitate flow and comprehension. 

In many instances, the paragraph starts talking about one topic and towards the end switches to another subject. The transitions could be made smoother so as to facilitate reading and avoid making the reader have to go back to previous paragraphs to fill in the gaps.”

I reviewed the article and tried to make the transitions smoother, as the reviewer suggested.

“For example: section 2.6 P1 (lines 215-223): talks about the various setups for MCG recordings. Last sentence then states that once systems became commercially available, "groups without cryogenic expertise were able to obtain clinical MCG data". A naive reader would ask: why is that? is this because such systems did not use cryogenic technology? or is it because those systems did not require the same level of maintenance - and hence knowledge- from users?”

I changed the final sentence to be “Furthermore, once commercial systems could be purchased off-the-shelf rather than being designed and manufactured in-house, groups without cryogenic expertise were able to obtain clinical MCG data.”

“Similarly, section 2.6 P2 (lines 224-229): talks about applications MCG but in the last sentence states that recently AI and ML have been used. This implies that the previously described analysis was manual in nature. The reader is also left wondering why this statement was made: why do we want to know that AI is now playing a role in data analysis? I'd recommend adding a sentence stating something like: ‘While the data analyses for all of these medical applications was originally done manually, recent advances in AI and ML have permitted a shift towards automatic MCG analysis to facilitate more rapid and accurate clinical diagnosis.’” 

I replaced my original sentence with “While the data analyses for all of these applications was originally performed using traditional methods, recent advances in artificial intelligence and machine learning have permitted a shift toward automatic MCG analysis to facilitate more rapid and accurate clinical diagnosis [38–40].” 

“Section 3.1 P3 (line 256-267): starts by talking about the purpose of experiments in measuring the magnetic field. But then switches the subjects to Toroids and their applications. The paragraph needs to be reorganized and a thesis sentence needs to be added. It could be rewritten as follows, for example: "Toroids have been used in a variety of experiments to measure the magnetic field from tissue." example 1, example 2, example 3.”

I added an introductory sentence to the paragraph that reads “Toroids have been used in a variety of experiments to measure the magnetic field from isolated tissue” and changed the wording in the rest of the paragraph so it provides a series of examples. 

“Section 3.2 last paragraph (line 302-311): talks about interpretation based on ampere's law. Then the last sentence talks about the electrical activity of the heart as a current dipole. A more obvious transition needs to be added so that the reader is not left wondering why we're talking about the heart or current dipoles. Transition phrases could help facilitate this ‘For example, in the case of...’.”  

I added “For example” at the start of the sentence.

"Several other paragraphs could benefit from having a thesis sentence at the beginning and more obvious transitions between sentences to facilitate reading and comprehension."

I reread the article and improved the transitions between sentences when possible, as suggested by the reviewer.

"Only minor editing is required (e.g unnecessary use of 'the')."

I removed uses of “the” when not necessary, as suggested by the reviewer.

Reviewer 3 Report

The current work focuses on Biomagnetism. The author’s great effort into the manuscript, but minor issues should be addressed.

-There is a felt lack of critical assessments by the authors. The authors did not mention the research gap between the previously reported articles and the present situation. The author should incorporate their views in each subsection to mold the research in a new direction.

 In the section “Clinical Applications” more details are required to write with the most recent related references.

- Conclusion and future perspective should write in detail.

Author Response

Reviewer 3:

I thank the reviewers for their comments, which have improved the manuscript. All changes in the manuscript are marked in red.

“The current work focuses on Biomagnetism. The author’s great effort into the manuscript, but minor issues should be addressed.

-There is a felt lack of critical assessments by the authors. The authors did not mention the research gap between the previously reported articles and the present situation. The author should incorporate their views in each subsection to mold the research in a new direction.”

I reviewed the article and added more critical assessment where possible, including a new paragraph at the send of Section 4.5 about the relative merits of MEG, EEG, fMRI, and PET.

“In the section 'Clinical Applications' more details are required to write with the most recent related references.”

I added additional details to the Clinical Applications section (Sec. 4.5).

"- Conclusion and future perspective should write in detail."

I didn’t change the conclusion, because I wanted to keep it short to avoid merely repeating statements from the text. However, I tried to emphasize conclusions and future perspectives in the text itself.